# The Role of Momentum Parameters in the Optimal Convergence of Adaptive Polyak's Heavy-ball Methods

**Wei Tao** [*]
Institute of Evaluation and Assessment Research
Academy of Military Science
Beijing, China
`wtao_plaust@163.com`

**Sheng Long** [*]
Department of Information Engineering
Army Academy of Artillery and Air Defense
Hefei, China
`ls15186322349@163.com`

**Gaowei Wu**
Institute of Automation CAS
School of Artificial Intelligence
University of CAS
Beijing, China
`gaowei.wu@ia.ac.cn`

**Qing Tao** [†]
Department of Information Engineering
Army Academy of Artillery and Air Defense
Hefei, China
`qing.tao@ia.ac.cn`

## Abstract

The adaptive stochastic gradient descent (SGD) with momentum has been widely adopted in deep learning as well as convex optimization. In practice, the last iterate is commonly used as the final solution. However, the available regret analysis and the setting of constant momentum parameters only guarantee the optimal convergence of the averaged solution. In this paper, we fill this theory-practice gap by investigating the convergence of the last iterate (referred to as *individual convergence*), which is a more difficult task than convergence analysis of the averaged solution. Specifically, in the constrained convex cases, we prove that the adaptive Polyak's Heavy-ball (HB) method, in which the step size is only updated using the exponential moving average strategy, attains an individual convergence rate of $O(\frac{1}{\sqrt{t}})$, as opposed to that of $O(\frac{\log t}{\sqrt{t}})$ of SGD, where $t$ is the number of iterations. Our new analysis not only shows how the HB momentum and its time-varying weight help us to achieve the acceleration in convex optimization but also gives valuable hints how the momentum parameters should be scheduled in deep learning. Empirical results validate the correctness of our convergence analysis in optimizing convex functions and demonstrate the improved performance of the adaptive HB methods in training deep networks.

## 1 Introduction

One of the most popular optimization algorithms in deep learning is the momentum method (Krizhevsky et al., 2012). The first momentum can be traced back to the pioneering work of Polyak's heavy-ball (HB) method (Polyak, 1964), which helps accelerate stochastic gradient descent (SGD) in the relevant direction and dampens oscillations (Ruder, 2016). Recent studies also find that the HB momentum has the potential to escape from the local minimum and saddle points (Ochs et al., 2014; Sun et al., 2019a). From the perspective of theoretical analysis, HB enjoys a smaller convergence factor than SGD when the objective function is twice continuously differentiable and strongly convex (Ghadimi et al., 2015). In nonsmooth convex cases, with suitably chosen step size, HB attains an optimal convergence rate of $O(\frac{1}{\sqrt{t}})$ in terms of the averaged output (Yang et al., 2016), where $t$ is the number of iterations.

---

[*]Equal contribution
[†]Corresponding author

To overcome the data-independent limitation of predetermined step size rules, some adaptive gradient methods have been proposed to exploit the geometry of historical data. The first algorithm in this line is AdaGrad (Duchi et al., 2011). The intuition behind AdaGrad is that the seldom-updated weights should be updated with a larger step size than the frequently-updated weights. Typically, AdaGrad rescales each coordinate and estimates the predetermined step size by a sum of squared past gradient values. As a result, AdaGrad has the same convergence rate as vanilla SGD but enjoys a smaller factor especially in sparse learning problems. The detailed analysis of AdaGrad (Mukkamala & Hein, 2017) implies that one can derive similar convergence rates for the adaptive variants of the predetermined step size methods without additional difficulties.

Unfortunately, experimental results illustrate that AdaGrad under-performed when applied to training deep neural newtworks (Wilson et al., 2017). Practical experience has led to the development of adaptive methods that is able to emphasize the more recent gradients. Specifically, an exponential moving average (EMA) strategy was proposed in RMSProp to replace the cumulative sum operation (Tieleman & Hinton, 2012). Adam (Kingma & Ba, 2014), which remains one of the most popular optimization algorithms in deep learning till today, built upon RMSProp together with updating the search directions via the HB momentum. Generally speaking, the gradient-based momentum algorithms that simultaneously update the search directions and learning rates using the past gradients are referred to as the Adam-type methods (Chen et al., 2019). These kinds of methods have achieved several state-of-the-art results on various learning tasks (Sutskever et al., 2013).

Compared with HB and AdaGrad, the main novelty of Adam lies in applying EMA to gradient estimate (first-order) and to element-wise square-of-gradients (second-order), with the momentum parameter $\beta_{1t}$ and step size parameter $\beta_{2t}$ (see (6)) (Alacaoglu et al., 2020). However, the use of EMA causes a lot of complexities to the convergence analysis. For example, in the online setting, (Kingma & Ba, 2014) offered a proof that Adam would converge to the optimum. Despite its remarkable practicality, Adam suffers from the non-convergence issue. To overcome its advantages, several variants such as AMSGrad and AdamNC were proposed (Reddi et al., 2018). Unfortunately, the regret bound of AMSGrad in (Reddi et al., 2018) is $O(\sqrt{\log t}\sqrt{t})$ for nonsmooth convex problems, as opposed to that of $O(\sqrt{t})$ of SGD. On the other hand, EMA uses the current step size in exponential moving averaging while the original HB can use the previous information (Zou et al., 2018). This will lead the update to stagnate when $\beta_{1t}$ is very close to 1. Fortunately, such a dilemma will not appear in Polyak's HB method and a simple proof on the convergence of this kind of Adams in smooth cases has been provided (Défossez et al., 2020).

In this paper, we will focus on the adaptive Polyak's HB method, in which the step size is only updated using EMA. Despite various reported practical performance for the Adam-type methods, there still exist some gaps between theoretical guarantees and empirical success.

- First of all, some important regret bounds have been established to guarantee the performance of online Adam-type algorithms. Nevertheless, the online-to-batch conversion can inevitably lead the solution of the induced stochastic algorithm to take the form of averaging of all the past iterates. In practice, the last iterate is popularly used as the final solution, which has the advantage of readily enforcing the learning structure (Chen et al., 2012). For SGD, the convergence of the last iterate, which is referred to as *individual convergence* in (Tao et al., 2020b), was posed as an open problem (Shamir, 2012). Only recently, its optimal individual convergence rate is proved to be $O(\frac{\log t}{\sqrt{t}})$ and $O(\frac{\log t}{t})$ for general and strongly convex problems respectively (Harvey et al., 2019; Jain et al., 2019). Despite enjoying the optimal averaging convergence (Yang et al., 2016), as far as we know, the individual convergence about the adaptive HB has not been discussed.

- Secondly, the momentum technique is often claimed as an accelerated strategy in machine learning community. However, almost all the theoretical analysis is only limited to the Nesterov's accelerated gradient (NAG) (Nesterov, 1983) method especially in smooth cases (Hu et al., 2009; Liu & Belkin, 2020), which accelerates the rate of SGD from $O(\frac{1}{t})$ to $O(\frac{1}{t^2})$. While the individual convergence of HB is also concerned in some papers (Sebbouh et al., 2020; Sun et al., 2019b), the considered problem is limited to smooth and the derived rate is not optimal in convex cases. It is discovered that NAG is capable of accelerating the rate of individual convergence of SGD from $O(\frac{\log t}{\sqrt{t}})$ to $O(\frac{1}{\sqrt{t}})$ (Tao et al.,

2020a) in nonsmooth convex cases. Nevertheless, there is still a lack of the report about the acceleration of the adaptive HB.

- Finally, in practice, almost all the momentum and Adam-type algorithms are often used with a constant momentum parameter $\beta_{1t}$ (typically between 0.9 and 0.99). In theory, regret guarantees in the online Adam require a rapidly decaying $\beta_{1t} \to 0$ schedule, which is also considered in (Sutskever et al., 2013; Orvieto et al., 2019). This gap is recently bridged by getting the same regret bounds as that in (Reddi et al., 2018) with a constant $\beta_{1t}$ (Alacaoglu et al., 2020). In each state-of-the-art deep learning library (e.g. TensorFlow, PyTorch, and Keras), HB is named as SGD with momentum and $\beta_{1t}$ is empirically set to 0.9 (Ruder, 2016). Despite its intuition in controlling the number of forgotten past gradients and guarantee in optimal averaging convergence (Yang et al., 2016), how $\beta_{1t}$ affects individual convergence has not been discussed (Gitman et al., 2019).

The goal of this paper is to close a theory-practice gap when using HB to train the deep neural networks as well as optimize the convex objective functions. Specifically,

- By setting $\beta_{1t} = \frac{t}{t+2}$, we prove that the adaptive HB attains an individual convergence rate of $O(\frac{1}{\sqrt{t}})$ (Theorem 5), as opposed to that of $O(\frac{\log t}{\sqrt{t}})$ of SGD. Our proof is different from all the existing analysis of averaging convergence. It not only provides a theoretical guarantee for the acceleration of HB but also clarifies how the momentum and its parameter $\beta_{1t}$ help us to achieve the optimal individual convergence.
- If $0 \le \beta_{1t} \equiv \beta < 1$, we prove that the adaptive HB attains optimal averaging convergence (Theorem 6). To guarantee the optimal individual convergence, Theorem 5 suggests that time-varying $\beta_{1t}$ can be adopted. Note $\beta_{1t} = \frac{t}{t+2} \to 1$, thus our new convergence analysis not only offers an interesting explanation why we usually restrict $\beta_{1t} \to 1$ but also gives valuable hints how the momentum parameters should be scheduled in deep learning.

We mainly focus on the proof of individual convergence of HB (Theorem 3, Appendix A.1). The analysis of averaging convergence (Theorem 4) is simpler. Their extensions to adaptive cases are slightly more complex (Theorem 5 and 6), but it is similar to the proof of AdaGrad (Mukkamala & Hein, 2017) and the details can be found in the supplementary material.

## 2 PROBLEM STATEMENT AND RELATED WORK

Consider the following optimization problem,

$$\min f(\mathbf{w}), \ s.t. \ \mathbf{w} \in \mathbf{Q}. \tag{1}$$

where $\mathbf{Q} \subseteq \mathbb{R}^d$ is a closed convex set and $f(\mathbf{w})$ is a convex function. Denote that $\mathbf{w}^*$ is an optimal solution and $P$ is the projection operator on $\mathbf{Q}$. Generally, the averaging convergence is defined as

$$f(\bar{\mathbf{w}}_t) - f(\mathbf{w}^*) \le \epsilon(t), \tag{2}$$

where $\bar{\mathbf{w}}_t = \frac{1}{t} \sum_{i=1}^{t} \mathbf{w}_i$ and $\epsilon(t)$ is the convergence bound about $t$. By contrast, the *individual convergence* is described as

$$f(\mathbf{w}_t) - f(\mathbf{w}^*) \le \epsilon(t). \tag{3}$$

Throughout this paper, we use $\mathbf{g}(\mathbf{w}_t)$ to denote the subgradient of $f$ at $\mathbf{w}_t$. Projected subgradient descent (PSG) is one of the most fundamental algorithms for solving problem (1) (Dimitri P. et al., 2003), and the iteration of which is

$$\mathbf{w}_{t+1} = P[\mathbf{w}_t - \alpha_t \mathbf{g}(\mathbf{w}_t)],$$

where $\alpha_t > 0$ is the step size. To analyze the convergence, we need the following assumption.

**Assumption 1**. *Assume that there exists a number $M > 0$ such that*

$$\|\mathbf{g}(\mathbf{w})\| \le M, \ \forall \mathbf{w} \in \mathbf{Q}.$$

It is known that the optimal bound for the nonsmooth convex problem (1) is $O(\frac{1}{\sqrt{t}})$ (Nemirovsky & Yudin, 1983). PSG can attain this optimal convergence rate in terms of the averaged output while its optimal individual rate is only $O(\frac{\log t}{\sqrt{t}})$ (Harvey et al., 2019; Jain et al., 2019).

When $\mathbf{Q} = \mathbb{R}^N$, the regular HB for solving the unconstrained problem (1) is

$$\mathbf{w}_{t+1} = \mathbf{w}_t - \alpha_t \mathbf{g}(\mathbf{w}_t) + \beta_t(\mathbf{w}_t - \mathbf{w}_{t-1}). \tag{4}$$

If $0 \leq \beta_t \equiv \beta < 1$, the key property of HB is that it can be reformulated as (Ghadimi et al., 2015)

$$\mathbf{w}_{t+1} + \mathbf{p}_{t+1} = \mathbf{w}_t + \mathbf{p}_t - \frac{\alpha_t}{1-\beta}\mathbf{g}(\mathbf{w}_t), \text{ where } \mathbf{p}_t = \frac{\beta}{1-\beta}(\mathbf{w}_t - \mathbf{w}_{t-1}). \tag{5}$$

Thus its convergence analysis makes almost no difference to that of PSG. Especially, if $\alpha_t \equiv \frac{\alpha}{\sqrt{T}}$, its averaging convergence rate is $O(\frac{1}{\sqrt{T}})$ (Yang et al., 2016), where $T$ is the total number of iterations.

Simply speaking, the regular Adam (Kingma & Ba, 2014) takes the form of

$$\mathbf{w}_{t+1} = \mathbf{w}_t - \frac{\alpha}{\sqrt{t}}V_t^{-\frac{1}{2}}\hat{\mathbf{g}}_t,$$

where $\hat{\mathbf{g}}(\mathbf{w}_t)$ is a unbiased estimation of $\mathbf{g}(\mathbf{w}_t)$ and

$$\hat{\mathbf{g}}_t = \beta_{1t}\hat{\mathbf{g}}_{t-1} + (1 - \beta_{1t})\hat{\mathbf{g}}(\mathbf{w}_t), \ V_t = \beta_{2t}V_{t-1} + (1 - \beta_{2t})\text{diag}\left[\hat{\mathbf{g}}(\mathbf{w}_t)\hat{\mathbf{g}}(\mathbf{w}_t)^\top\right]. \tag{6}$$

## 3 INDIVIDUAL CONVERGENCE OF HB

To solve the constrained problem (1), HB can be naturally reformulated as

$$\mathbf{w}_{t+1} = P_{\mathbf{Q}}[\mathbf{w}_t - \alpha_t \mathbf{g}(\mathbf{w}_t) + \beta_t(\mathbf{w}_t - \mathbf{w}_{t-1})]. \tag{7}$$

We first prove a key lemma, which extends (5) to the constrained and time-varying cases.

**Lemma 1.** *(Dimitri P. et al., 2003) For $\mathbf{w} \in \mathbb{R}^d, \mathbf{w}_0 \in \mathbf{Q}$,*

$$\langle \mathbf{w} - \mathbf{w}_0, \mathbf{u} - \mathbf{w}_0 \rangle \leq 0,$$

*for all $\mathbf{u} \in \mathbf{Q}$ if and only if $\mathbf{w}_0 = P(\mathbf{w})$.*

**Lemma 2.** *Let $\{\mathbf{w}_t\}_{t=1}^{\infty}$ be generated by HB (7). Let*

$$\mathbf{p}_t = t(\mathbf{w}_t - \mathbf{w}_{t-1}), \ \beta_t = \frac{t}{t+2}, \ \alpha_t = \frac{\alpha}{(t+2)\sqrt{t}}.$$

*Then HB (7) can be reformulated as*

$$\mathbf{w}_{t+1} + \mathbf{p}_{t+1} = P_{\mathbf{Q}}[\mathbf{w}_t + \mathbf{p}_t - \frac{\alpha}{\sqrt{t}}\mathbf{g}(\mathbf{w}_t)]. \tag{8}$$

**Proof**. The projection operation can be rewritten as an optimization problem (Duchi, 2018), i.e., $\mathbf{w}_{t+1} = P_{\mathbf{Q}}[\mathbf{w}_t - \alpha_t \mathbf{g}(\mathbf{w}_t) + \beta_t(\mathbf{w}_t - \mathbf{w}_{t-1})]$ is equivalent to

$$\mathbf{w}_{t+1} = \arg\min_{\mathbf{w} \in \mathbf{Q}}\{\alpha_t\langle\mathbf{g}(\mathbf{w}_t), \mathbf{w}\rangle + \frac{1}{2}\|\mathbf{w} - \mathbf{w}_t - \beta_t(\mathbf{w}_t - \mathbf{w}_{t-1})\|^2\}. \tag{9}$$

Then, $\forall \mathbf{w} \in \mathbf{Q}$, we have

$$\langle\mathbf{w}_{t+1} - \mathbf{w}_t - \beta_t(\mathbf{w}_t - \mathbf{w}_{t-1}) + \alpha_t\mathbf{g}(\mathbf{w}_t), \mathbf{w}_{t+1} - \mathbf{w}\rangle \leq 0.$$

This is

$$\langle\mathbf{w}_{t+1} + \mathbf{p}_{t+1} - (\mathbf{w}_t + \mathbf{p}_t) + \frac{\alpha}{\sqrt{t}}\mathbf{g}(\mathbf{w}_t), \mathbf{w}_{t+1} - \mathbf{w}\rangle \leq 0. \tag{10}$$

Specifically,

$$\langle\mathbf{w}_{t+1} + \mathbf{p}_{t+1} - (\mathbf{w}_t + \mathbf{p}_t) + \frac{\alpha}{\sqrt{t}}\mathbf{g}(\mathbf{w}_t), \mathbf{w}_{t+1} - \mathbf{w}_t\rangle \leq 0. \tag{11}$$

From (10) and (11),

$$\langle\mathbf{w}_{t+1} + \mathbf{p}_{t+1} - (\mathbf{w}_t + \mathbf{p}_t) + \frac{\alpha}{\sqrt{t}}\mathbf{g}(\mathbf{w}_t), \mathbf{w}_{t+1} - \mathbf{w} + (t+1)(\mathbf{w}_{t+1} - \mathbf{w_t})\rangle \leq 0.$$

i.e.,

$$\langle\mathbf{w}_{t+1} + \mathbf{p}_{t+1} - (\mathbf{w}_t + \mathbf{p}_t) + \frac{\alpha}{\sqrt{t}}\mathbf{g}(\mathbf{w}_t), \mathbf{w}_{t+1} + \mathbf{p}_{t+1} - \mathbf{w}\rangle \leq 0.$$

Using Lemma 1, Lemma 2 is proved.

Due to the non-expansive property of $P_{\mathbf{Q}}$ (Dimitri P. et al., 2003), Lemma 2 implies that the convergence analysis for unconstrained problems can be applied to analyze the constrained problems.

**Theorem 3.** *Assume that* $\mathbf{Q}$ *is bounded. Let* $\{\mathbf{w}_t\}_{t=1}^{\infty}$ *be generated by HB (7). Set*

$$\beta_t = \frac{t}{t+2} \text{ and } \alpha_t = \frac{\alpha}{(t+2)\sqrt{t}}.$$

*Then*

$$f(\mathbf{w}_t) - f(\mathbf{w}^*) \leq O(\frac{1}{\sqrt{t}}).$$

**Proof.** According to Lemma 2,

$$\|\mathbf{w}^* - (\mathbf{w}_{t+1} + \mathbf{p}_{t+1})\|^2 \leq \|\mathbf{w}^* - (\mathbf{w}_t + \mathbf{p}_t) + \frac{\alpha}{\sqrt{t}}\mathbf{g}(\mathbf{w}_t)\|^2.$$

$$\|\mathbf{w}^* - (\mathbf{w}_t + \mathbf{p}_t) + \frac{\alpha}{\sqrt{t}}\mathbf{g}(\mathbf{w}_t)\|^2$$

$$= \|\mathbf{w}^* - (\mathbf{w}_t + \mathbf{p}_t)\|^2 + \|\frac{\alpha}{\sqrt{t}}\mathbf{g}(\mathbf{w}_t)\|^2 + 2\langle\frac{\alpha}{\sqrt{t}}\mathbf{g}(\mathbf{w}_t), \mathbf{w}^* - \mathbf{w}_t\rangle + 2\langle\frac{\alpha t}{\sqrt{t}}\mathbf{g}(\mathbf{w}_t), \mathbf{w}_{t-1} - \mathbf{w}_t\rangle$$

Note

$$\langle\mathbf{g}(\mathbf{w}_t), \mathbf{w}^* - \mathbf{w}_t\rangle \leq f(\mathbf{w}^*) - f(\mathbf{w}_t), \ \langle\mathbf{g}(\mathbf{w}_t), \mathbf{w}_{t-1} - \mathbf{w}_t\rangle \leq f(\mathbf{w}_{t-1}) - f(\mathbf{w}_t).$$

Then

$$(t+1)[f(\mathbf{w}_t) - f(\mathbf{w}^*)]$$

$$\leq t[f(\mathbf{w}_{t-1}) - f(\mathbf{w}^*)] + \frac{\sqrt{t}}{2\alpha}\|\mathbf{w}^* - (\mathbf{w}_t + \mathbf{p}_t)\|^2 - \frac{\sqrt{t}}{2\alpha}\|\mathbf{w}^* - (\mathbf{w}_{t+1} + \mathbf{p}_{t+1})\|^2 + \frac{\alpha}{2\sqrt{t}}\|\mathbf{g}(\mathbf{w}_t)\|^2.$$

Summing this inequality from $k = 1$ to $t$, we obtain

$$(t+1)[f(\mathbf{w}_t) - f(\mathbf{w}^*)]$$

$$\leq f(\mathbf{w}_0) - f(\mathbf{w}^*) + \sum_{k=1}^{t}\frac{\alpha}{2\sqrt{k}}\|\mathbf{g}(\mathbf{w}_k)\|^2 + \sum_{k=1}^{t}\left[\frac{\sqrt{k}}{2\alpha}(\|\mathbf{w}^* - (\mathbf{w}_k + \mathbf{p}_k)\|^2 - \|\mathbf{w}^* - (\mathbf{w}_{k+1} + \mathbf{p}_{k+1})\|^2)\right].$$

Note

$$\sum_{k=1}^{t}\frac{1}{2\sqrt{k}}\|\mathbf{g}(\mathbf{w}_k)\|^2 \leq \sqrt{t}M^2.$$

and

$$\sum_{k=1}^{t}\left[\frac{\sqrt{k}}{2}(\|\mathbf{w}^* - (\mathbf{w}_k + \mathbf{p}_k)\|^2 - \|\mathbf{w}^* - (\mathbf{w}_{k+1} + \mathbf{p}_{k+1})\|^2)\right].$$

$$\leq \frac{1}{2}\|\mathbf{w}^* - (\mathbf{w}_1 + \mathbf{p}_1)\|^2 - \frac{\sqrt{t}}{2}\|\mathbf{w}^* - (\mathbf{w}_{t+1} + \mathbf{p}_{t+1})\|^2 + \sum_{k=2}^{t}(\frac{\sqrt{k}}{2} - \frac{\sqrt{k-1}}{2})\|\mathbf{w}^* - (\mathbf{w}_k + \mathbf{p}_k)\|^2.$$

Since $\mathbf{Q}$ is a bounded set, there exists a positive number $M_0 > 0$ such that

$$\|\mathbf{w}^* - (\mathbf{w}_{t+1} + \mathbf{p}_{t+1})\|^2 \leq M_0, \forall t \geq 0.$$

Therefore

$$(t+1)[f(\mathbf{w}_t) - f(\mathbf{w}^*)] \leq f(\mathbf{w}_0) - f(\mathbf{w}^*) + \alpha\sqrt{t}M^2 + \frac{\sqrt{t}}{2\alpha}M_0.$$

This completes the proof of Theorem 3.

It is necessary to give some remarks about Theorem 3.

- In nonsmooth convex cases, Theorem 3 shows that the individual convergence rate of SGD can be accelerated from $O(\frac{\log t}{\sqrt{t}})$ to $O(\frac{1}{\sqrt{t}})$ via the HB momentum. The proof here clarifies how the HB-type momentum $\mathbf{w}_t - \mathbf{w}_{t-1}$ and its time-varying weight $\beta_t$ help us to derive the optimal individual convergence.

- The convergence analysis in Theorem 3 is obviously different from the regret analysis in all the available papers, this is because the connection between $f(\mathbf{w}_t) - f(\mathbf{w}^*)$ and $f(\mathbf{w}_{t-1}) - f(\mathbf{w}^*)$ should be established here. It can be seen that seeking an optimal individual convergence is more difficult than the analysis of averaging convergence in many papers such as (Zinkevich, 2003) and (Yang et al., 2016).

- We can get a stochastic HB by replacing the subgradient $\mathbf{g}(\mathbf{w}_t)$ in (7) with its unbiased estimation $\hat{\mathbf{g}}(\mathbf{w}_t)$. Such substitution will not influence our convergence analysis. This means that we can get $\mathbb{E}[f(\mathbf{w}_t) - f(\mathbf{w}^*)] \leq O(\frac{1}{\sqrt{t}})$ under the same assumptions.

If $\beta_t$ remains a constant, we can get the averaging convergence rate, in which the proof of the first part is similar to Lemma 2 and that of the second part is similar to online PSG (Zinkevich, 2003).

**Theorem 4.** *Assume that $\mathbf{Q}$ is bounded and $0 \leq \beta_t \equiv \beta < 1$. Let $\{\mathbf{w}_t\}_{t=1}^{\infty}$ be generated by HB (7). Set*

$$\mathbf{p}_t = \frac{\beta}{1-\beta}(\mathbf{w}_t - \mathbf{w}_{t-1}) \text{ and } \alpha_t = \frac{\alpha}{\sqrt{t}}.$$

*Then we have*

$$\mathbf{w}_{t+1} + \mathbf{p}_{t+1} = P_\mathbf{Q}[\mathbf{w}_t + \mathbf{p}_t - \frac{\alpha_t}{1-\beta}\mathbf{g}(\mathbf{w}_t)], \ f(\frac{1}{t}\sum_{k=1}^{t}\mathbf{w}_k) - f(\mathbf{w}^*) \leq O(\frac{1}{\sqrt{t}}).$$

If $\mathbf{Q}$ is not bounded, the boundness of sequence $\|\mathbf{w}^* - (\mathbf{w}_{t+1} + \mathbf{p}_{t+1})\|$ can not be ensured, which may lead to the failure of Theorem 4. Fortunately, like that in (Yang et al., 2016), $\mathbb{E}[f(\frac{1}{T}\sum_{k=1}^{T}\mathbf{w}_k) - f(\mathbf{w}^*)] \leq O(\frac{1}{\sqrt{T}})$ still holds, but we need to set $\alpha_t \equiv \frac{\alpha}{\sqrt{T}}$.

## 4 EXTENSION TO ADAPTIVE CASES

It is easy to find that HB (8) is in fact a gradient-based algorithm with predetermined step size $\frac{\alpha}{\sqrt{t}}$. Thus its adaptive variant with EMA can be naturally formulated as

$$\mathbf{w}_{t+1} = P_\mathbf{Q}[\mathbf{w}_t - \frac{\alpha\beta_{1t}}{t\sqrt{t}}V_t^{-\frac{1}{2}}\hat{\mathbf{g}}(\mathbf{w}_t) + \beta_{1t}(\mathbf{w}_t - \mathbf{w}_{t-1})]. \tag{12}$$

where

$$\beta_{1t} = \frac{t}{t+2}, \ V_t = \beta_{2t}V_{t-1} + (1-\beta_{2t})\text{diag}\left[\hat{\mathbf{g}}(\mathbf{w}_t)\hat{\mathbf{g}}(\mathbf{w}_t)^\top\right].$$

The detailed steps of the adaptive HB are shown in Algorithm 1.

---

**Algorithm 1** Adaptive HB

---

**Input:** momentum parameters $\beta_{1t}, \beta_{2t}$, constant $\delta > 0$, the total number of iterations $T$
1: Initialize $\mathbf{w}_0 = \mathbf{0}, V_0 = \mathbf{0}_{d \times d}$
2: **repeat**
3:     $\hat{\mathbf{g}}_t(\mathbf{w}_t) = \nabla f_t(\mathbf{w}_t)$,
4:     $V_t = \beta_{2t}V_{t-1} + (1-\beta_{2t})diag(\hat{\mathbf{g}}_t(\mathbf{w}_t)\hat{\mathbf{g}}_t(\mathbf{w}_t)^\top)$,
5:     $\hat{V}_t = V_t^{\frac{1}{2}} + \frac{\delta}{\sqrt{t}}I_d$,
6:     $\mathbf{w}_{t+1} = P_\mathbf{Q}[\mathbf{w}_t - \frac{\alpha\beta_{1t}}{t\sqrt{t}}\hat{V}_t^{-1}\hat{\mathbf{g}}(\mathbf{w}_t) + \beta_{1t}(\mathbf{w}_t - \mathbf{w}_{t-1})]$,
7: **until** $t = T$
**Output:** $\mathbf{w}_T$

---

**Theorem 5.** *Assume that $\mathbf{Q}$ is a bounded set. Let $\{\mathbf{w}_t\}_{t=1}^{\infty}$ be generated by the adaptive HB (Algorithm 1). Denote $\mathbf{p}_t = t(\mathbf{w}_t - \mathbf{w}_{t-1})$. Suppose that $\beta_{1t} = \frac{t}{t+2}$ and $1 - \frac{1}{t} \leq \beta_{2t} \leq 1 - \frac{\gamma}{t}$ for some $0 < \gamma \leq 1$. Then*

$$\mathbf{w}_{t+1} + \mathbf{p}_{t+1} = P_\mathbf{Q}[\mathbf{w}_t + \mathbf{p}_t - \frac{\alpha}{\sqrt{t}}\hat{V}_t^{-1}\hat{\mathbf{g}}(\mathbf{w}_t)] \tag{13}$$

$$\mathbb{E}[f(\mathbf{w}_t) - f(\mathbf{w}^*)] \leq O(\frac{1}{\sqrt{t}}).$$

The proof of (13) is identical to that of Lemma 2. It is easy to find that (13) is an adaptive variant of (8). This implies that the proof of the second part is similar to that of AdaGrad (Mukkamala & Hein, 2017). When $0 \leq \beta_{1t} \equiv \beta < 1$, the adaptive variant of HB (7) is

$$\mathbf{w}_{t+1} = P_{\mathbf{Q}}[\mathbf{w}_t - \frac{\alpha}{\sqrt{t}}V_t^{-\frac{1}{2}}\hat{\mathbf{g}}(\mathbf{w}_t) + \beta(\mathbf{w}_t - \mathbf{w}_{t-1})].  \tag{14}$$

where

$$V_t = \beta_{2t}V_{t-1} + (1 - \beta_{2t})\mathrm{diag}(\hat{\mathbf{g}}(\mathbf{w}_t)\hat{\mathbf{g}}(\mathbf{w}_t)^{\top}).$$

Similar to the proof of Theorem 5, we can get the following averaging convergence.

**Theorem 6.** *Assume that* $\mathbf{Q}$ *is bounded and* $0 \leq \beta_{1t} \equiv \beta < 1$ *in Algorithm 1. Let* $\{\mathbf{w}_t\}_{t=1}^{\infty}$ *be generated by the adaptive HB (Algorithm 1). Suppose that* $1 - \frac{1}{t} \leq \beta_{2t} \leq 1 - \frac{\gamma}{t}$ *for some* $0 < \gamma \leq 1$. *Denote* $\mathbf{p}_t = \frac{\beta}{1-\beta}(\mathbf{w}_t - \mathbf{w}_{t-1})$. *Then*

$$\mathbf{w}_{t+1} + \mathbf{p}_{t+1} = P_{\mathbf{Q}}[\mathbf{w}_t + \mathbf{p}_t - \frac{\alpha}{(1-\beta)\sqrt{t}}\hat{V}_t^{-1}\hat{\mathbf{g}}(\mathbf{w}_t)], \ \mathbb{E}[f(\frac{1}{t}\sum_{k=1}^{t}\mathbf{w}_k) - f(\mathbf{w}^*)] \leq O(\frac{1}{\sqrt{t}}).$$

It is necessary to give some remarks about Theorem 5 and Theorem 6.

- The adaptive HB is usually used with a constant $\beta_{1t}$ in deep learning. However, according to Theorem 6, the constant $\beta_{1t}$ only guarantees the optimal data-dependent averaging convergence. The convergence property of the last iterate still remains unknown.

- In order to assure the optimal individual convergence, according to Theorem 5, $\beta_{1t}$ has to be time-varying. $\beta_{1t} = \frac{t}{t+2}$ can explain why we usually restrict $\beta_{1t} \to 1$ in practice. It also offers a new schedule about the selection of momentum parameters in deep learning.

## 5 EXPERIMENTS

In this section, we present some empirical results. The first two experiments are to validate the correctness of our convergence analysis and investigate the performance of the suggested parameters schedule. For fair comparison, we independently repeat the experiments five times and report the averaged results. The last experiment (Appendix A.4) is to show the effective acceleration of HB over GD in terms of the individual convergence.

### 5.1 EXPERIMENTS ON OPTIMIZING GENERAL CONVEX FUNCTIONS

This experiment is to optimize hinge loss with the $l_1$-ball constraints. Let $\tau$ denotes the radius of the $l_1$-ball. For implementation of the $l_1$ projection operation, we use SLEP package[1].

$$\min f(\mathbf{w}), \ s.t. \ \mathbf{w} \in \{\mathbf{w} : \|\mathbf{w}\|_1 \leq \tau\}.  \tag{15}$$

Datasets: A9a, W8a, Covtype, Ijcnn1, Rcv1, Realsim (available at LibSVM[2] website).

Algorithms: PSG ($\alpha_t = \frac{\alpha}{\sqrt{t}}$), HB ($\alpha_t = \frac{\alpha}{(t+2)\sqrt{t}}$, $\beta_t = \frac{t}{t+2}$), NAG (Tao et al., 2020a) and adaptive HB (12) ($\beta_{1t} = \frac{t}{t+2}$).

The relative function value $f(\mathbf{w}_t) - f(\mathbf{w}_*)$ v.s. epoch is illustrated in Figure 1. As expected, the individual convergence of the adaptive HB has almost the same behavior as the averaging output of PSG, and the individual output of HB and NAG. Since the three stochastic methods have the optimal convergence for general convex problems, we conclude that the stochastic adaptive HB attains the optimal individual convergence.

---

[1] http://yelabs.net/software/SLEP/
[2] http://www.csie.ntu.edu.tw/~cjlin/libsvmtools/datasets/

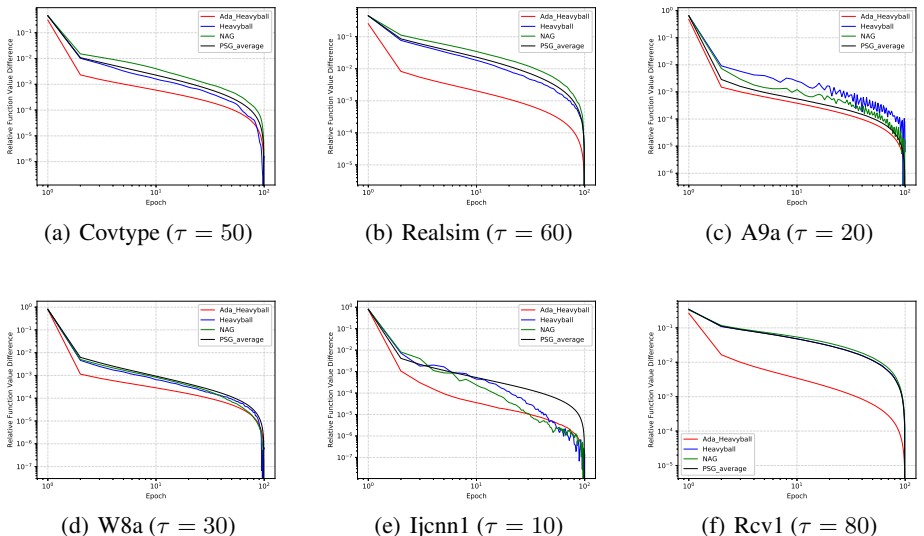

Figure 1: Convergence on different LibSVM datasets

## 5.2 TRAINING DEEP NEURAL NETWORKS

These experiments are conducted on 4-layer CNN and ResNet-18 using a server with 2 NVIDIA 2080Ti GPUs.

Datasets: MNIST (60000 training samples, 10000 test samples), CIFAR10 (50000 training samples, 10000 test samples), and CIFAR100 (50000 training samples, 10000 test samples).

Algorithms: Adam ($\alpha$, $\beta_{1t} \equiv 0.9$, $\beta_{2t} \equiv 0.999$, $\epsilon = 10^{-8}$) (Kingma & Ba, 2014), SGD ($\alpha_t \equiv \alpha$), SGD-momentum ($\alpha_t \equiv \alpha$, $\beta_t \equiv 0.9$), AdaGrad ($\alpha_t \equiv \alpha$) (Duchi et al., 2011), RMSprop ($\alpha_t \equiv \alpha$, $\beta_{2t} \equiv 0.9$, $\epsilon = 10^{-8}$) (Tieleman & Hinton, 2012). For our adaptive HB, $\gamma = 0.1$ and $\delta = 10^{-8}$. Different from the existing methods, we set $\beta_{1t} = \frac{t}{t+2}$ and $\beta_{2t} = 1 - \frac{\gamma}{t}$ in Algorithm 1. Within each epoch, $\beta_{1t}$ and $\beta_{2t}$ remain unchanged.

Note that all methods have only one adjustable parameter $\alpha$, we choose $\alpha$ from the set of $\{0.1, 0.01, 0.001, 0.0001\}$ for all experiments. Following (Mukkamala & Hein, 2017) and (Wang et al., 2020), we design a simple 4-layer CNN architecture that consists two convolutional layers (32 filters of size $3 \times 3$), one max-pooling layer ($2 \times 2$ window and 0.25 dropout) and one fully connected layer (128 hidden units and 0.5 dropout). We also use weight decay and batch normalization to reduce over-fitting. The optimal rate is always chosen for each algorithm separately so that one achieves either best training objective or best test performance after a fixed number of epochs.

The loss function is the cross-entropy. The training loss results are illustrated in Figure 2 and 4, and the test accuracy results are presented in Figure 3 and 5. As can be seen, the adaptive HB achieves the improved training loss. Moreover, this improvement also leads to good performance on test accuracy. The experimental results show that our suggested schedule about the momentum parameters could gain improved practical performance even in deep learning tasks.

## 6 CONCLUSION

In this paper, we prove that the adaptive HB method attains an optimal data-dependent individual convergence rate in the constrained convex cases, which bridges a theory-practice gap in using momentum methods to train the deep neural networks as well as optimize the convex functions. Our new analysis not only clarifies how the HB momentum and its time-varying weight $\beta_{1t} = \frac{t}{t+2}$ help us to achieve the acceleration but also gives valuable hints how its momentum parameters should be scheduled in deep learning. Empirical results on optimizing convex functions validate the

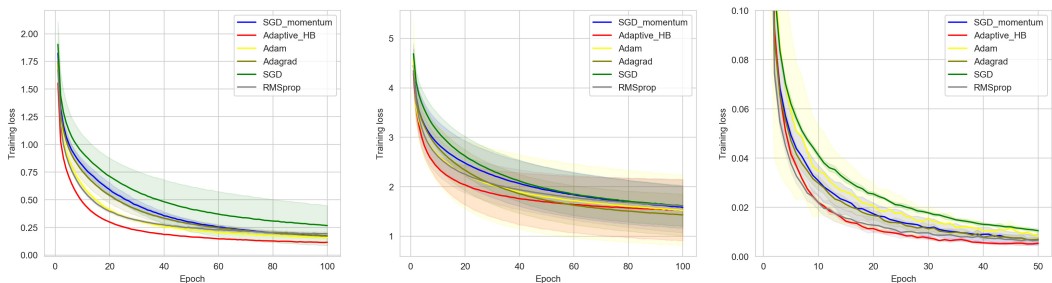

Figure 2: Training loss v.s. number of epochs on 4-layer CNN: CIFAR10, CIFAR100, MNIST

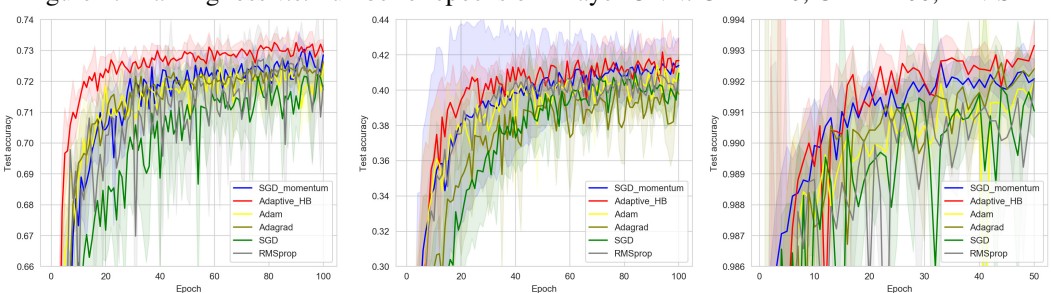

Figure 3: Test accuracy v.s. number of epochs on 4-layer CNN: CIFAR10, CIFAR100, MNIST

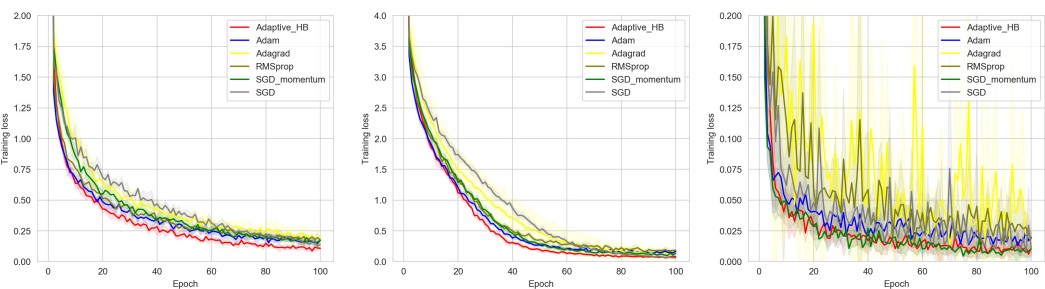

Figure 4: Training loss v.s. number of epochs on ResNet-18: CIFAR10, CIFAR100, MNIST

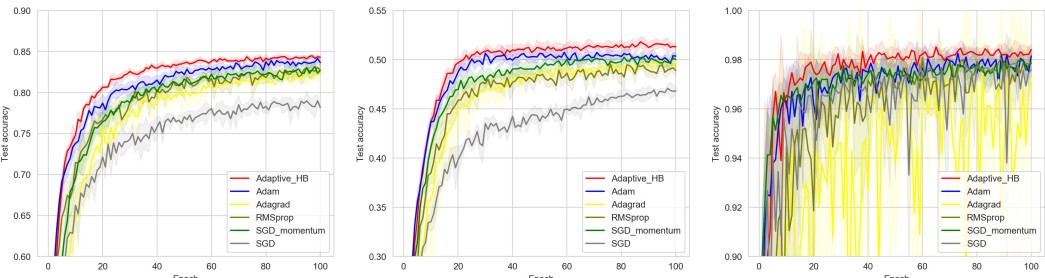

Figure 5: Test accuracy v.s. number of epochs on ResNet-18: CIFAR10, CIFAR100, MNIST

correctness of our convergence analysis and several typical deep learning experiments demonstrate the improved performance of the adaptive HB.

# 7 ACKNOWLEDGEMENTS

This work was supported in part by National Natural Science Foundation of China under Grants (62076252, 61673394, 61976213) and in part by Beijing Advanced Discipline Fund.

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

## A   SUPPLEMENTARY MATERIAL

### A.1   PROOF FOR THEOREM 4

Let $\{\mathbf{w}_t\}_{t=1}^{\infty}$ be generated by HB (7). Set

$$\mathbf{p}_t = \frac{\beta}{1-\beta}(\mathbf{w}_t - \mathbf{w}_{t-1}) \text{ and } \alpha_t = \frac{\alpha}{\sqrt{t}}.$$

Then, $\forall \mathbf{w} \in \mathbf{Q}$, according to Lemma 1, we have

$$\langle \mathbf{w}_{t+1} - \mathbf{w}_t - \beta(\mathbf{w}_t - \mathbf{w}_{t-1}) + \alpha_t \mathbf{g}(\mathbf{w}_t), \mathbf{w}_{t+1} - \mathbf{w} \rangle \leq 0.$$

This is

$$\langle \frac{1}{1-\beta}(\mathbf{w}_{t+1} - \mathbf{w}_t) - \mathbf{p}_t + \frac{\alpha_t}{1-\beta}\mathbf{g}(\mathbf{w}_t), \mathbf{w}_{t+1} - \mathbf{w} \rangle \leq 0.$$

i.e.,

$$\langle \mathbf{w}_{t+1} + \mathbf{p}_{t+1} - (\mathbf{w}_t + \mathbf{p}_t) + \frac{\alpha_t}{1-\beta}\mathbf{g}(\mathbf{w}_t), \mathbf{w}_{t+1} - \mathbf{w} \rangle \leq 0 \tag{16}$$

Specifically,

$$\langle \mathbf{w}_{t+1} + \mathbf{p}_{t+1} - (\mathbf{w}_t + \mathbf{p}_t) + \frac{\alpha_t}{1-\beta}\mathbf{g}(\mathbf{w}_t), \frac{\beta(\mathbf{w}_{t+1} - \mathbf{w}_t)}{1-\beta} \rangle \leq 0 \tag{17}$$

From (16) and (17),

$$\langle \mathbf{w}_{t+1} + \mathbf{p}_{t+1} - (\mathbf{w}_t + \mathbf{p}_t) + \frac{\alpha_t}{1-\beta}\mathbf{g}(\mathbf{w}_t), \mathbf{w}_{t+1} + \mathbf{p}_{t+1} - \mathbf{w} \rangle \leq 0.$$

Using Lemma 1, we have

$$\mathbf{w}_{t+1} + \mathbf{p}_{t+1} = P_{\mathbf{Q}}[\mathbf{w}_t + \mathbf{p}_t - \frac{\alpha_t}{1-\beta}\mathbf{g}(\mathbf{w}_t)].$$

Then

$$\|\mathbf{w}^* - (\mathbf{w}_{t+1} + \mathbf{p}_{t+1})\|^2$$

$$\leq \|\mathbf{w}^* - (\mathbf{w}_t + \mathbf{p}_t) + \frac{\alpha_t}{1-\beta}\mathbf{g}(\mathbf{w}_t)\|^2$$

$$= \|\mathbf{w}^* - (\mathbf{w}_t + \mathbf{p}_t)\|^2 + \|\frac{\alpha_t}{1-\beta}\mathbf{g}(\mathbf{w}_t)\|^2 + 2\langle \frac{\alpha_t}{1-\beta}\mathbf{g}(\mathbf{w}_t), \mathbf{w}^* - \mathbf{w}_t \rangle$$

$$+ 2\langle \frac{\alpha_t \beta}{(1-\beta)^2}\mathbf{g}(\mathbf{w}_t), \mathbf{w}_{t-1} - \mathbf{w}_t \rangle$$

Note

$$\langle \mathbf{g}(\mathbf{w}_t), \mathbf{w}^* - \mathbf{w}_t \rangle \leq f(\mathbf{w}^*) - f(\mathbf{w}_t), \quad \langle \mathbf{g}(\mathbf{w}_t), \mathbf{w}_{t-1} - \mathbf{w}_t \rangle \leq f(\mathbf{w}_{t-1}) - f(\mathbf{w}_t).$$

Then

$$\|\mathbf{w}^* - (\mathbf{w}_{t+1} + \mathbf{p}_{t+1})\|^2$$

$$\leq \|\mathbf{w}^* - (\mathbf{w}_t + \mathbf{p}_t)\|^2 + \frac{\alpha_t^2}{(1-\beta)^2}\|\mathbf{g}(\mathbf{w}_t)\|^2$$

$$+ \frac{2\alpha_t}{1-\beta}[f(\mathbf{w}^*) - f(\mathbf{w}_t)] + \frac{2\alpha_t \beta}{(1-\beta)^2}[f(\mathbf{w}_{t-1}) - f(\mathbf{w}_t)].$$

Rearrange the inequality, we have

$$\frac{2\alpha_t}{1-\beta}[f(\mathbf{w}_t) - f(\mathbf{w}^*)] \leq \frac{2\alpha_t \beta}{(1-\beta)^2}[f(\mathbf{w}_{t-1}) - f(\mathbf{w}_t)] + \|\mathbf{w}^* - (\mathbf{w}_t + \mathbf{p}_t)\|^2$$

$$- \|\mathbf{w}^* - (\mathbf{w}_{t+1} + \mathbf{p}_{t+1})\|^2 + \frac{\alpha_t^2}{(1-\beta)^2}\|\mathbf{g}(\mathbf{w}_t)\|^2.$$

i.e.,

$$f(\mathbf{w}_t) - f(\mathbf{w}^*) \le \frac{\beta}{1-\beta}[f(\mathbf{w}_{t-1}) - f(\mathbf{w}_t)] + \frac{1-\beta}{2\alpha_t}[\|\mathbf{w}^* - (\mathbf{w}_t + \mathbf{p}_t)\|^2$$
$$- \|\mathbf{w}^* - (\mathbf{w}_{t+1} + \mathbf{p}_{t+1})\|^2] + \frac{\alpha_t}{2(1-\beta)}\|\mathbf{g}(\mathbf{w}_t)\|^2.$$

Summing this inequality from $k = 1$ to $t$, we obtain

$$\sum_{k=1}^{t}[f(\mathbf{w}_k) - f(\mathbf{w}^*)]$$
$$\le \frac{\beta}{1-\beta}[f(\mathbf{w}_0) - f(\mathbf{w}_t)] + \frac{1-\beta}{2\alpha_1}\|\mathbf{w}^* - (\mathbf{w}_1 + \mathbf{p}_1)\|^2$$
$$- \frac{1-\beta}{2\alpha_t}\|\mathbf{w}^* - (\mathbf{w}_{t+1} + \mathbf{p}_{t+1})\|^2 + \sum_{k=1}^{t}\frac{\alpha_k}{2(1-\beta)}\|\mathbf{g}(\mathbf{w}_k)\|^2$$
$$+ \sum_{k=2}^{t}\|\mathbf{w}^* - (\mathbf{w}_k + \mathbf{p}_k)\|^2(\frac{1-\beta}{2\alpha_k} - \frac{1-\beta}{2\alpha_{k-1}}).$$

i.e.,

$$\sum_{k=1}^{t}[f(\mathbf{w}_k) - f(\mathbf{w}^*)]$$
$$\le \frac{\beta}{1-\beta}[f(\mathbf{w}_0) - f(\mathbf{w}_t)] + \frac{1-\beta}{2\alpha}\|\mathbf{w}^* - (\mathbf{w}_1 + \mathbf{p}_1)\|^2$$
$$+ \sum_{k=2}^{t}\|\mathbf{w}^* - (\mathbf{w}_k + \mathbf{p}_k)\|^2(\frac{(1-\beta)\sqrt{k}}{2\alpha} - \frac{(1-\beta)\sqrt{k-1}}{2\alpha}) \qquad (18)$$
$$+ \sum_{k=1}^{t}\frac{\alpha}{2(1-\beta)\sqrt{k}}\|\mathbf{g}(\mathbf{w}_k)\|^2.$$

Note

$$\sum_{k=1}^{t}\frac{1}{2\sqrt{k}}\|\mathbf{g}(\mathbf{w}_k)\|^2 \le \sqrt{t}M^2. \qquad (19)$$

and since $\mathbf{Q}$ is a bounded set, there exists a positive number $M_0 > 0$ such that

$$\|\mathbf{w}^* - (\mathbf{w}_{t+1} + \mathbf{p}_{t+1})\|^2 \le M_0, \forall t \ge 0. \qquad (20)$$

From (18)(19)(20) we have,

$$\sum_{k=1}^{t}[f(\mathbf{w}_k) - f(\mathbf{w}^*)] \le \frac{\beta}{1-\beta}[f(\mathbf{w}_0) - f(\mathbf{w}_t)] + \frac{(1-\beta)\sqrt{t}M_0}{2\alpha} + \frac{\alpha\sqrt{t}M^2}{1-\beta}.$$

By convexity of $f(\mathbf{w})$, we obtain

$$f(\frac{1}{t}\sum_{k=1}^{t}\mathbf{w}_k) - f(\mathbf{w}^*) \le \frac{\beta}{(1-\beta)t}[f(\mathbf{w}_0) - f(\mathbf{w}_t)] + \frac{(1-\beta)M_0}{2\alpha\sqrt{t}} + \frac{\alpha M^2}{(1-\beta)\sqrt{t}}.$$

This completes the proof of Theorem 4.

### A.2 PROOF FOR THEOREM 5

Notation. For a positive definite matrix $H \in \mathbb{R}^{d\times d}$, the weighted $\ell_2$-norm is defined by $\|\mathbf{x}\|_H^2 = \mathbf{x}^\top H\mathbf{x}$. The $H$-weighted projection $P_{\mathbf{Q}}^H(\mathbf{x})$ of $\mathbf{x}$ onto $\mathbf{Q}$ is defined by $P_{\mathbf{Q}}^H(\mathbf{x}) = \arg\min_{\mathbf{y}\in\mathbf{Q}}\|\mathbf{y} - \mathbf{x}\|_H^2$. We use $\mathbf{g}(\mathbf{w}_k)$ to denote the subgradient of $f_k(\cdot)$ at $\mathbf{w}_k$. For the diagonal matrix sequence $\{M_k\}_{k=1}^t$, we use $m_{k,i}$ to denote the $i$-th element in the diagonal of $M_k$. We introduce the notation, $g_{1:k,i} = (g_{1,i}, g_{2,i}, .., g_{k,i})^\top$, where $g_{k,i}$ is the $i$-th element of $\mathbf{g}(\mathbf{w}_k)$.

**Lemma 7.** *(Mukkamala & Hein, 2017) Suppose that* $1 - \frac{1}{t} \leq \beta_{2t} \leq 1 - \frac{\gamma}{t}$ *for some* $0 < \gamma \leq 1$, *and* $t \geq 1$, *then*

$$\sum_{i=1}^{d} \sum_{k=1}^{t} \frac{g_{k,i}^2}{\sqrt{k v_{k,i}} + \delta} \leq \sum_{i=1}^{d} \frac{2(2-\gamma)}{\gamma} (\sqrt{t v_{t,i}} + \delta).$$

**Proof for Theorem 5.** Without loss of generality, we only prove Theorem 5 in the full gradient setting. It can be extended to stochastic cases using the regular technique in (Rakhlin et al., 2011).

Note that the projection operation can be rewritten as an optimization problem (Duchi, 2018), i.e., $\mathbf{w}_{t+1} = P_{\mathbf{Q}}[\mathbf{w}_t - \alpha_t \hat{V}_t^{-1} \mathbf{g}(\mathbf{w}_t) + \beta_{1t}(\mathbf{w}_t - \mathbf{w}_{t-1})]$ is equivalent to

$$\mathbf{w}_{t+1} = \arg\min_{\mathbf{w} \in \mathbf{Q}} \{ \alpha_t \langle \hat{V}_t^{-1} \mathbf{g}(\mathbf{w}_t), \mathbf{w} \rangle + \frac{1}{2} \|\mathbf{w} - \mathbf{w}_t - \beta_{1t}(\mathbf{w}_t - \mathbf{w}_{t-1})\|^2 \}. \tag{21}$$

Then, $\forall \mathbf{u} \in \mathbf{Q}$, we have

$$\langle \mathbf{w}_{t+1} - \mathbf{w}_t - \beta_t(\mathbf{w}_t - \mathbf{w}_{t-1}) + \alpha_t \hat{V}_t^{-1} \mathbf{g}(\mathbf{w}_t), \mathbf{w}_{t+1} - \mathbf{w} \rangle \leq 0.$$

This is

$$\langle \mathbf{w}_{t+1} + \mathbf{p}_{t+1} - (\mathbf{w}_t + \mathbf{p}_t) + \frac{\alpha}{\sqrt{t}} \hat{V}_t^{-1} \mathbf{g}(\mathbf{w}_t), \mathbf{w}_{t+1} - \mathbf{w} \rangle \leq 0. \tag{22}$$

Specifically,

$$\langle \mathbf{w}_{t+1} + \mathbf{p}_{t+1} - (\mathbf{w}_t + \mathbf{p}_t) + \frac{\alpha}{\sqrt{t}} \hat{V}_t^{-1} \mathbf{g}(\mathbf{w}_t), \mathbf{w}_{t+1} - \mathbf{w}_t \rangle \leq 0. \tag{23}$$

From (22) and (23),

$$\langle \mathbf{w}_{t+1} + \mathbf{p}_{t+1} - (\mathbf{w}_t + \mathbf{p}_t) + \frac{\alpha}{\sqrt{t}} \hat{V}_t^{-1} \mathbf{g}(\mathbf{w}_t), \mathbf{w}_{t+1} - \mathbf{w}_t + (t+1)(\mathbf{w}_{t+1} - \mathbf{w}_t) \rangle \leq 0.$$

i.e.,

$$\langle \mathbf{w}_{t+1} + \mathbf{p}_{t+1} - (\mathbf{w}_t + \mathbf{p}_t) + \frac{\alpha}{\sqrt{t}} \hat{V}_t^{-1} \mathbf{g}(\mathbf{w}_t), \mathbf{w}_{t+1} + \mathbf{p}_{t+1} - \mathbf{w}_t \rangle \leq 0.$$

Using Lemma 1, we have

$$\mathbf{w}_{t+1} + \mathbf{p}_{t+1} = P_{\mathbf{Q}}^{\hat{V}_t}[\mathbf{w}_t + \mathbf{p}_t - \frac{\alpha}{\sqrt{t}} \hat{V}_t^{-1} \mathbf{g}(\mathbf{w}_t)].$$

Then

$$\|\mathbf{w}^* - (\mathbf{w}_{t+1} + \mathbf{p}_{t+1})\|_{\hat{V}_t}^2 \leq \|\mathbf{w}^* - (\mathbf{w}_t + \mathbf{p}_t) + \frac{\alpha}{\sqrt{t}} \hat{V}_t^{-1} \mathbf{g}(\mathbf{w}_t)\|_{\hat{V}_t}^2$$

$$= \|\mathbf{w}^* - (\mathbf{w}_t + \mathbf{p}_t)\|_{\hat{V}_t}^2 + \|\frac{\alpha}{\sqrt{t}} \mathbf{g}(\mathbf{w}_t)\|_{\hat{V}_t}^2$$

$$+ 2\langle \frac{\alpha}{\sqrt{t}} \mathbf{g}(\mathbf{w}_t), \mathbf{w}^* - \mathbf{w}_t \rangle + 2\langle \frac{\alpha t}{\sqrt{t}} \mathbf{g}(\mathbf{w}_t), \mathbf{w}_{t-1} - \mathbf{w}_t \rangle.$$

Note

$$\langle \mathbf{g}(\mathbf{w}_t), \mathbf{w}^* - \mathbf{w}_t \rangle \leq f(\mathbf{w}^*) - f(\mathbf{w}_t), \quad \langle \mathbf{g}(\mathbf{w}_t), \mathbf{w}_{t-1} - \mathbf{w}_t \rangle \leq f(\mathbf{w}_{t-1}) - f(\mathbf{w}_t).$$

Then

$$(t+1)[f(\mathbf{w}_t) - f(\mathbf{w}^*)] \leq t[f(\mathbf{w}_{t-1}) - f(\mathbf{w}^*)] + \frac{\sqrt{t}}{2\alpha} \|\mathbf{w}^* - (\mathbf{w}_t + \mathbf{p}_t)\|_{\hat{V}_t}^2$$

$$- \frac{\sqrt{t}}{2\alpha} \|\mathbf{w}^* - (\mathbf{w}_{t+1} + \mathbf{p}_{t+1})\|_{\hat{V}_t}^2 + \frac{\alpha}{2\sqrt{t}} \|\mathbf{g}(\mathbf{w}_t)\|_{\hat{V}_t^{-1}}^2.$$

Summing this inequality from $k = 1$ to $t$, we obtain

$$(t+1)[f(\mathbf{w}_t) - f(\mathbf{w}^*)] \leq f(\mathbf{w}_0) - f(\mathbf{w}^*) + \sum_{k=1}^{t} \frac{\alpha}{2\sqrt{k}} \|\mathbf{g}(\mathbf{w}_k)\|_{\hat{V}_k^{-1}}^2$$

$$+ \sum_{k=1}^{t} \left[ \frac{\sqrt{k}}{2\alpha} (\|\mathbf{w}^* - (\mathbf{w}_k + \mathbf{p}_k)\|_{\hat{V}_k}^2 - \|\mathbf{w}^* - (\mathbf{w}_{k+1} + \mathbf{p}_{k+1})\|_{\hat{V}_k}^2) \right].$$

Using Lemma 7, we have

$$\sum_{k=1}^{t} \frac{\alpha}{2\sqrt{k}} \|\mathbf{g}(\mathbf{w}_k)\|^2_{\hat{V}_k^{-1}} \le \sum_{i=1}^{d} \frac{\alpha(2-\gamma)}{\gamma}(\sqrt{tv_{t,i}} + \delta).$$

Note

$$\sum_{k=1}^{t} \left[ \frac{\sqrt{k}}{2\alpha}(\|\mathbf{w}^* - (\mathbf{w}_k + \mathbf{p}_k)\|^2_{\hat{V}_k} - \|\mathbf{w}^* - (\mathbf{w}_{k+1} + \mathbf{p}_{k+1})\|^2_{\hat{V}_k}) \right]$$

$$= \sum_{i=1}^{d} \frac{\hat{v}_{1,i}}{2\alpha}(\mathbf{w}_i^* - (\mathbf{w}_{1,i} + \mathbf{p}_{1,i}))^2 - \sum_{i=1}^{d} \frac{\sqrt{t}\hat{v}_{t,i}}{2\alpha}(\mathbf{w}_i^* - (\mathbf{w}_{t+1,i} + \mathbf{p}_{t+1,i}))^2 \qquad (24)$$

$$+ \sum_{i=1}^{d} \sum_{k=2}^{t} \frac{1}{2\alpha}(\sqrt{k}\hat{v}_{k,i} - \sqrt{k-1}\hat{v}_{k-1,i})(\mathbf{w}_i^* - (\mathbf{w}_{k,i} + \mathbf{p}_{k,i}))^2.$$

Since $\mathbf{Q}$ is a bounded set, there exists a positive number $M_1 > 0$ such that

$$(\mathbf{w}_i^* - (\mathbf{w}_{t+1,i} + \mathbf{p}_{t+1,i}))^2 \le M_1, \forall t \ge 0, i = 1, 2, ..., d.$$

and $v_{k,i} = \beta_{2k}v_{k-1,i} + (1 - \beta_{2k})g^2_{k,i}$ as well as $\beta_{2k} \ge 1 - \frac{1}{k}$ which implies $k\beta_{2k} \ge k - 1$, we get

$$\sqrt{k}\hat{v}_{k,i} = \sqrt{kv_{k,i}} + \delta$$

$$= \sqrt{k\beta_{2k}v_{k-1,i} + k(1 - \beta_{2k})g^2_{k,i}} + \delta$$

$$\ge \sqrt{(k-1)v_{k-1,i}} + \delta$$

$$= \sqrt{k-1}\hat{v}_{k-1,i}.$$

Therefore

$$\sum_{k=1}^{t} \left[ \frac{\sqrt{k}}{2\alpha}(\|\mathbf{w}^* - (\mathbf{w}_k + \mathbf{p}_k)\|^2_{\hat{V}_k} - \|\mathbf{w}^* - (\mathbf{w}_{k+1} + \mathbf{p}_{k+1})\|^2_{\hat{V}_k}) \right]$$

$$\le \sum_{i=1}^{d} \frac{\hat{v}_{1,i}}{2\alpha}M_1 + \sum_{i=1}^{d} \sum_{k=2}^{t} \frac{1}{2\alpha}(\sqrt{k}\hat{v}_{k,i} - \sqrt{k-1}\hat{v}_{k-1,i})M_1$$

$$= \sum_{i=1}^{d} \frac{\hat{v}_{1,i}M_1}{2\alpha} + \sum_{i=1}^{d} \frac{\sqrt{t}\hat{v}_{t,i}M_1}{2\alpha} - \sum_{i=1}^{d} \frac{\hat{v}_{1,i}M_1}{2\alpha} \qquad (25)$$

$$= \frac{M_1}{2\alpha} \sum_{i=1}^{d} (\sqrt{tv_{t,i}} + \delta).$$

Since $\sqrt{tv_{t,i}} = \|g_{1:t,i}\|$, therefore

$$(t+1)[f(\mathbf{w}_t) - f(\mathbf{w}^*)] \le f(\mathbf{w}_0) - f(\mathbf{w}^*) + \frac{M_1}{2\alpha} \sum_{i=1}^{d}(\sqrt{tv_{t,i}} + \delta) + \sum_{i=1}^{d} \frac{\alpha(2-\gamma)}{\gamma}(\sqrt{tv_{t,i}} + \delta)$$

$$= f(\mathbf{w}_0) - f(\mathbf{w}^*) + (\frac{M_1}{2\alpha} + \frac{\alpha(2-\gamma)}{\gamma}) \sum_{i=1}^{d}(\|g_{1:t,i}\| + \delta).$$

This proves

$$f(\mathbf{w}_t) - f(\mathbf{w}^*) \le O(\frac{1}{\sqrt{t}}).$$

### A.3 PROOF FOR THEOREM 6

Let $\{\mathbf{w}_t\}_{t=1}^{\infty}$ be generated by the adaptive HB (Algorithm 1). Set

$$\mathbf{p}_t = \frac{\beta}{1-\beta}(\mathbf{w}_t - \mathbf{w}_{t-1}) \text{ and } \alpha_t = \frac{\alpha}{\sqrt{t}}.$$

Then, $\forall \mathbf{u} \in \mathbf{Q}$, according to Lemma 1, we have

$$\langle \mathbf{w}_{t+1} - \mathbf{w}_t - \beta(\mathbf{w}_t - \mathbf{w}_{t-1}) + \alpha_t \hat{V}_t^{-1} \mathbf{g}(\mathbf{w}_t), \mathbf{w}_{t+1} - \mathbf{w} \rangle \leq 0.$$

This is

$$\langle \frac{1}{1-\beta}(\mathbf{w}_{t+1} - \mathbf{w}_t) - \mathbf{p}_t + \frac{\alpha_t \hat{V}_t^{-1}}{1-\beta} \mathbf{g}(\mathbf{w}_t), \mathbf{w}_{t+1} - \mathbf{w} \rangle \leq 0.$$

i.e.,

$$\langle \mathbf{w}_{t+1} + \mathbf{p}_{t+1} - (\mathbf{w}_t + \mathbf{p}_t) + \frac{\alpha_t \hat{V}_t^{-1}}{1-\beta} \mathbf{g}(\mathbf{w}_t), \mathbf{w}_{t+1} - \mathbf{w} \rangle \leq 0 \tag{26}$$

Specifically,

$$\langle \mathbf{w}_{t+1} + \mathbf{p}_{t+1} - (\mathbf{w}_t + \mathbf{p}_t) + \frac{\alpha_t \hat{V}_t^{-1}}{1-\beta} \mathbf{g}(\mathbf{w}_t), \frac{\beta(\mathbf{w}_{t+1} - \mathbf{w}_t)}{1-\beta} \rangle \leq 0 \tag{27}$$

From (26) and (27),

$$\langle \mathbf{w}_{t+1} + \mathbf{p}_{t+1} - (\mathbf{w}_t + \mathbf{p}_t) + \frac{\alpha_t \hat{V}_t^{-1}}{1-\beta} \mathbf{g}(\mathbf{w}_t), \mathbf{w}_{t+1} + \mathbf{p}_{t+1} - \mathbf{w} \rangle \leq 0.$$

Using Lemma 1, we have

$$\mathbf{w}_{t+1} + \mathbf{p}_{t+1} = P_{\mathbf{Q}}^{\hat{V}_t}[\mathbf{w}_t + \mathbf{p}_t - \frac{\alpha_t \hat{V}_t^{-1}}{1-\beta} \mathbf{g}(\mathbf{w}_t)].$$

According to Lemma 2,

$$\|\mathbf{w}^* - (\mathbf{w}_{t+1} + \mathbf{p}_{t+1})\|_{\hat{V}_t}^2$$

$$\leq \|\mathbf{w}^* - (\mathbf{w}_t + \mathbf{p}_t) + \frac{\alpha_t \hat{V}_t^{-1}}{1-\beta} \mathbf{g}(\mathbf{w}_t)\|_{\hat{V}_t}^2$$

$$= \|\mathbf{w}^* - (\mathbf{w}_t + \mathbf{p}_t)\|_{\hat{V}_t}^2 + \|\frac{\alpha_t}{1-\beta} \mathbf{g}(\mathbf{w}_t)\|_{\hat{V}_t}^2$$

$$+ 2\langle \frac{\alpha_t}{1-\beta} \mathbf{g}(\mathbf{w}_t), \mathbf{w}^* - \mathbf{w}_t \rangle + 2\langle \frac{\alpha_t \beta}{(1-\beta)^2} \mathbf{g}(\mathbf{w}_t), \mathbf{w}_{t-1} - \mathbf{w}_t \rangle$$

Note

$$\langle \mathbf{g}(\mathbf{w}_t), \mathbf{w}^* - \mathbf{w}_t \rangle \leq f(\mathbf{w}^*) - f(\mathbf{w}_t), \ \langle \mathbf{g}(\mathbf{w}_t), \mathbf{w}_{t-1} - \mathbf{w}_t \rangle \leq f(\mathbf{w}_{t-1}) - f(\mathbf{w}_t).$$

Then

$$\|\mathbf{w}^* - (\mathbf{w}_{t+1} + \mathbf{p}_{t+1})\|_{\hat{V}_t}^2$$

$$\leq \|\mathbf{w}^* - (\mathbf{w}_t + \mathbf{p}_t)\|_{\hat{V}_t}^2 + \frac{\alpha_t^2}{(1-\beta)^2} \|\mathbf{g}(\mathbf{w}_t)\|_{\hat{V}_t^{-1}}^2$$

$$+ \frac{2\alpha_t}{1-\beta}[f(\mathbf{w}^*) - f(\mathbf{w}_t)] + \frac{2\alpha_t \beta}{(1-\beta)^2}[f(\mathbf{w}_{t-1}) - f(\mathbf{w}_t)].$$

Rearrange the inequality, we have

$$\frac{2\alpha_t}{1-\beta}[f(\mathbf{w}_t) - f(\mathbf{w}^*)]$$

$$\leq \frac{2\alpha_t \beta}{(1-\beta)^2}[f(\mathbf{w}_{t-1}) - f(\mathbf{w}_t)] + \|\mathbf{w}^* - (\mathbf{w}_t + \mathbf{p}_t)\|_{\hat{V}_t}^2$$

$$- \|\mathbf{w}^* - (\mathbf{w}_{t+1} + \mathbf{p}_{t+1})\|_{\hat{V}_t}^2 + \frac{\alpha_t^2}{(1-\beta)^2} \|\mathbf{g}(\mathbf{w}_t)\|_{\hat{V}_t^{-1}}^2.$$

i.e.,

$$f(\mathbf{w}_t) - f(\mathbf{w}^*)$$

$$\leq \frac{\beta}{1-\beta}[f(\mathbf{w}_{t-1}) - f(\mathbf{w}_t)] + \frac{1-\beta}{2\alpha_t}[\|\mathbf{w}^* - (\mathbf{w}_t + \mathbf{p}_t)\|_{\hat{V}_t}^2$$

$$- \|\mathbf{w}^* - (\mathbf{w}_{t+1} + \mathbf{p}_{t+1})\|_{\hat{V}_t}^2] + \frac{\alpha_t}{2(1-\beta)} \|\mathbf{g}(\mathbf{w}_t)\|_{\hat{V}_t^{-1}}^2.$$

Summing this inequality from $k = 1$ to $t$, we obtain

$$
\sum_{k=1}^{t}[f(\mathbf{w}_k) - f(\mathbf{w}^*)]
$$
$$
\leq \frac{\beta}{1-\beta}[f(\mathbf{w}_0) - f(\mathbf{w}_t)] + \frac{1-\beta}{2\alpha_1}\|\mathbf{w}^* - (\mathbf{w}_1 + \mathbf{p}_1)\|_{\hat{V}_1}^2
$$
$$
- \frac{1-\beta}{2\alpha_t}\|\mathbf{w}^* - (\mathbf{w}_{t+1} + \mathbf{p}_{t+1})\|_{\hat{V}_t}^2 + \sum_{k=1}^{t}\frac{\alpha_k}{2(1-\beta)}\|\mathbf{g}(\mathbf{w}_k)\|_{\hat{V}_k^{-1}}^2
$$
$$
+ \sum_{i=1}^{d}\sum_{k=2}^{t}(\mathbf{w}_i^* - (\mathbf{w}_{k,i} + \mathbf{p}_{k,i}))^2 \left(\frac{(1-\beta)\hat{v}_{k,i}}{2\alpha_k} - \frac{(1-\beta)\hat{v}_{k-1,i}}{2\alpha_{k-1}}\right).
$$

i.e.,

$$
\sum_{k=1}^{t}[f(\mathbf{w}_k) - f(\mathbf{w}^*)]
$$
$$
\leq \frac{\beta}{1-\beta}[f(\mathbf{w}_0) - f(\mathbf{w}_t)] + \frac{1-\beta}{2\alpha}\|\mathbf{w}^* - (\mathbf{w}_1 + \mathbf{p}_1)\|_{\hat{V}_1}^2
$$
$$
+ \sum_{i=1}^{d}\sum_{k=2}^{t}(\mathbf{w}_i^* - (\mathbf{w}_{k,i} + \mathbf{p}_{k,i}))^2 \frac{1-\beta}{2\alpha}(\sqrt{k}\hat{v}_{k,i} - \sqrt{k-1}\hat{v}_{k-1,i}) \tag{28}
$$
$$
+ \sum_{k=1}^{t}\frac{\alpha}{2(1-\beta)\sqrt{k}}\|\mathbf{g}(\mathbf{w}_k)\|_{\hat{V}_k^{-1}}^2.
$$

Using Lemma 7, we have

$$
\sum_{k=1}^{t}\frac{\alpha}{2\sqrt{k}(1-\beta)}\|\mathbf{g}(\mathbf{w}_k)\|_{\hat{V}_k^{-1}}^2 \leq \sum_{i=1}^{d}\frac{\alpha(2-\gamma)}{\gamma(1-\beta)}(\sqrt{tv_{t,i}} + \delta) = \frac{\alpha(2-\gamma)}{\gamma(1-\beta)}\sum_{i=1}^{d}(\|g_{1:t,i}\| + \delta). \tag{29}
$$

and since $\mathbf{Q}$ is a bounded set, there exists a positive number $M_0 > 0$ such that

$$
\|\mathbf{w}^* - (\mathbf{w}_{t+1} + \mathbf{p}_{t+1})\|^2 \leq M_0, \forall t \geq 0. \tag{30}
$$

From (28)(29)(30) we have,

$$
\sum_{k=1}^{t}[f(\mathbf{w}_k) - f(\mathbf{w}^*)]
$$
$$
\leq \frac{\beta}{1-\beta}[f(\mathbf{w}_0) - f(\mathbf{w}_t)] + \sum_{i=1}^{d}\frac{(1-\beta)\hat{v}_{1,i}M_0}{2\alpha} + \frac{\alpha(2-\gamma)}{\gamma(1-\beta)}\sum_{i=1}^{d}(\|g_{1:t,i}\| + \delta)
$$
$$
+ \sum_{i=1}^{d}\frac{(1-\beta)\hat{v}_{t,i}\sqrt{t}M_0}{2\alpha} - \sum_{i=1}^{d}\frac{(1-\beta)\hat{v}_{1,i}M_0}{2\alpha}.
$$

i.e.,

$$
\sum_{k=1}^{t}[f(\mathbf{w}_k) - f(\mathbf{w}^*)] \leq \frac{\beta}{1-\beta}[f(\mathbf{w}_0) - f(\mathbf{w}_t)] + \frac{\alpha(2-\gamma)}{\gamma(1-\beta)}\sum_{i=1}^{d}(\|g_{1:t,i}\| + \delta) + \frac{(1-\beta)M_0}{2\alpha}\sum_{i=1}^{d}(\|g_{1:t,i}\| + \delta).
$$

By convexity of $f(\mathbf{w}_k)$, we obtain

$$
f(\frac{1}{t}\sum_{k=1}^{t}\mathbf{w}_k) - f(\mathbf{w}^*) \leq \frac{\beta}{(1-\beta)t}[f(\mathbf{w}_0) - f(\mathbf{w}_t)] + \frac{\alpha(2-\gamma)}{\gamma(1-\beta)t}\sum_{i=1}^{d}(\|g_{1:t,i}\| + \delta) + \frac{(1-\beta)M_0}{2\alpha t}\sum_{i=1}^{d}(\|g_{1:t,i}\| + \delta).
$$

This completes the proof of Theorem 6.

### A.4 EXPERIMENTS ON OPTIMIZING A SYNTHETIC CONVEX FUNCTION

A constrained convex optimization problem was constructed in (Harvey et al., 2019) to show that the optimal individual convergence rate of SGD is $O(\frac{\log t}{\sqrt{t}})$. We will use example to illustrate the acceleration of HB.

Let $\mathbf{Q}$ be unit ball in $\mathbb{R}^T$. For $i \in [T]$ and $c \geq 1$, define the positive scalar parameters

$$a_i = \frac{1}{8c(T-i+1)} \qquad b_i = \frac{\sqrt{i}}{2c\sqrt{T}}$$

Define $f : \mathbf{Q} \to \mathbb{R}$ and $\mathbf{h}_i \in \mathbb{R}^T$ for $i \in [T+1]$ by

$$f(\mathbf{w}) = \max_{i \in [T]} \mathbf{h}_i^\top \mathbf{w} \qquad where \qquad h_{i,j} = \begin{cases} a_j, & 1 \leq j < i \\ -b_j, & i = j < T \\ 0, & i < j \leq T \end{cases}$$

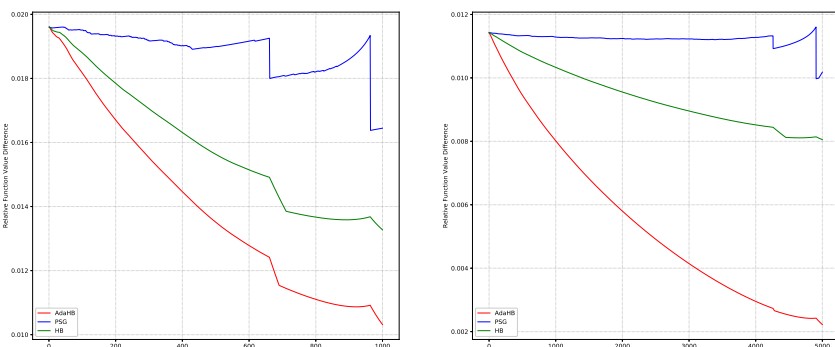

Figure 6: Convergence of the function value when $T = 1000$ and $T = 5000$

Obviously, the minimum value of $f$ on the unit ball is non-positive because $f(0) = 0$. It can be proved $f(\mathbf{w}_T) \geq \frac{\log T}{32c\sqrt{T}}$. Set $c = 2$, the function value $f(\mathbf{w}_t)$ v.s. iteration is illustrated in Figure 6, where the step size of GD is $\frac{c}{\sqrt{t}}$ and the parameters of the constrained HB (7) ($\alpha = 8$) and AdaHB (12) ($\alpha = 0.08$, $\gamma = 0.9$, $\delta = 10^{-8}$) are selected according to Theorem 3 and Theorem 5. As expected, the individual convergence of HB is much faster than that of PSG. We thus conclude that HB is an effective acceleration of GD in terms of the individual convergence.

