# OpenReview forum: "The Role of Momentum Parameters in the Optimal Convergence of Adaptive Polyak's Heavy-ball Methods"
_ICLR.cc/2021/Conference — ICLR 2021 Poster_

### Official Review · AnonReviewer2 · 2020-10-27

**Rating:** 6
**Confidence:** 3

**Review:**

Summary:

This paper provides the convergence analysis of the Heavy-ball method with the individual convergence and provides the convergence of its adaptive version.

Pros:

1. The motivation is interesting, most of time we use the last iteration while most of theorem can only provide convergence result for the average output. I am glad to see that the theoretical proof of the individual convergence for stochastic Heavy-ball method is studied in this paper,

2. The convergence result of HB methods achieved in this paper is 1/\sqrt(t) which is better the optimal result of SGD. This demonstrate the advantage of momentum-based methods.

3. Their proof is different form all the existing analysis of averaging convergence.

Cons:

1. It will be much better if the author can explain more on the difference between the current proof and the existing proof.


Overall, I tend to accept this paper. However I am not an expert on this area and I will not be sad if this paper is rejected by others.

---

> ### Author Response · Authors · 2020-11-17
> **Response to R2**
>
> Thank you for your efforts reviewing our paper and providing helpful comments and suggestions.
>
> Pros:
>
> A: Thanks for your positive evaluation.
>
> Cons:
>
> A: The main novelty of our individual convergence analysis is that we employ the momentum term to establish a recursion between $f(w_t) - f(w)$ and $f(w_{t-1}) - f(w)$, in which the time-varying momentum parameters are suitably selected. The existing proof mainly concerned about the bound of regret, in which only $f(w_t) - f(w)$ is required to consider.

---

### Official Review · AnonReviewer4 · 2020-10-28
**This work study the the individual convergence of Polyak momentum method in solving non-convex problems.**

**Rating:** 6
**Confidence:** 4

**Review:**

Summary:

The convergence of Momentum methods has been widely studied but most of existing works consider the average convergence. In this work the author considers the individual convergence of the last iteration. The accelerated convergence rate O(1/sqrt(t)) is established.

Pros:
1. The individual convergence of momentum methods is pretty interesting. It is very valuable to established the accelerated convergence rate, which is missing in the existing litterature.

2. The paper overall is well written but need to reword some statement for rigorousness.

Cons:
1. This paper only consider convex optimization. This limits its effectiveness to explain the good performance of momentum method in training deep neural networks.

2. I suggest the authors to reword their statement of the third contribution on page 3 "However, in order to get the optimal individual convergence, β1t has to be time-varying". Theorem 4 only shows that adaptive momentum can achieve individual convergence. Theorem 5 shows that constant momentum can achieve average convergence. These two results cannot lead to the conclusion that constant moment does not have individual convergence. In fact, we have observed it in practice. Moreover, the author should add more discussion on the difficulty of establish constant convergence for constant momentum

3. There is a large gap between the numerical and theoretical results. In the theoretical result, the authors consider convex optimization which is quite different from training deep neural networks. Moreover, when training DNN, the algorithm used is SGD with momentum but not GD with momentum. Note that M-SGD and M-GD has significantly difference, I am not sure how the theoretical results can provide useful hints to practice.

Minor Comments:
1. The abstract mentions that the numerical result on convex optimization is included. However, I don't find this result. Please add it.
2. I hope the authors can provide more proof sketch to help the reader understand the proof.

---

> ### Author Response · Authors · 2020-11-17
> **Response to R4**
>
> Thank you for your efforts reviewing our paper and providing helpful comments and suggestions.
>
> Cons:
>
> A1: Like many papers, we only focus on the theoretical analysis for convex problems.
>
> BTW: (1) Almost all the state-of-art optimizers including SGD and Adam also suffer from this problem. (2) Even if the nonconvex problems are considered, their convergence analysis inevitably needs some assumptions that actually does not hold in deep learning.
>
> A2: GD is a specific example of HB with $\beta_{1t}=0$, and it has been proved that GD cannot attain the optimal individual convergence rate of O(1/t) (Harvey et al., 2019). Based on this, we said that the constant momentum cannot guarantee the optimal individual convergence. The statement has been rephrased as “To guarantee the optimal individual convergence, Theorem 5 suggests that time-varying $\beta_{1t}$ can be adopted.”
>
> In fact, establishing averaging convergence for constant momentum is easier than analyzing the individual convergence due to no needing to select the time-varying parameters.
>
> A3: Following many papers, such as (Wang et al., 2020), we only focus on the theoretical analysis for convex problems. Indeed, the large gap exists for almost all the optimizers in deep learning. So far, there have been some papers considering nonconvex problems. However, their analysis still needs many additional assumptions that actually does not hold in deep learning.
>
> The theoretical results about general stochastic and adaptive settings have been provided in the original supplementary material.
>
> In experiments, we set $\beta_{1t}=\frac{t}{t+2}$ in Algorithm 1, which is motivated by the convergence analysis for convex problems and different from the existing methods.
>
> Minor Comments:
>
> A1: It has been included in the original supplementary material. We have also updated our paper with a revision to extend the Appendix.
>
> A2: Thanks for your advice. To help the reader understand the proof, only concise part is given in the paper. The complicated settings are provided in the supplementary material.

---

### Official Review · AnonReviewer3 · 2020-10-28
**An interesting read that still requires a bit of work**

**Rating:** 6
**Confidence:** 4

**Review:**

The authors investigate the convergence of the projected Heavy-ball method (and an adaptive variant) for convex problems with convex constraints. The authors prove 4 results: 2 individual (last iterate) convergence rates and 2 rates using averaging. Notably, in their proofs they require an increasing (from 1/2 to 1) momentum parameter and a decreasing stepsize. Finally, the authors present some experimental results.

The paper is overall a nice and pleasant read, with no major typos and a nice introduction. Also, to the best of my knowledge, the individual convergence rates proved by the authors are novel.
However, I do not think the paper is ready for publication, for the following reasons (in order of importance, the last points are easy-to-fix)

1) The paper is short – the main section is only 2 pages and the proof (in the appendix) does not exceed one page. I am not particularly blown away by the results or the proof technique – I think the authors should try to extend their study. For instance, an easy way to make the result more general is to extend it to momentum parameters of the form t/(t+r) for r>2 (it should work the same).

2) Reading the paper, it seems the authors are the first to provide an individual convergence rate for HB. This is not true, indeed Ghadimi 2015 also has a rate of O(1/t) for the last iterate of HB under momentum t/(t+2)  and stepsize 1/t. Here, the authors add a projection, hence have to also reduce the stepsize to 1/t^3/2.. @authors is this the only novelty? Also, an increasing momentum is considered in Orvieto et al. 2019 (Role of memory in stochastic optimization).

3) Even though the authors claim to have convex empirical results in the abstract, the experiments are for non convex problems (a CNN model). This in my view does not make much sense since (a) the authors only study convex problems with convex constraints (where are the constraints here?), (b) no experiment has a decreasing stepsize (c) a momentum of k/(k+2) was shown already in Sutskever et al. 2013 not to be optimal in the non convex case. In essence, I think the experiments do not back up the theory: the authors should at least compare with a fixed-momentum method.

4) I am seriously not convinced that O(1/t) compared to O(log(t)/t ) motivates the so-called “acceleration” of Heavy ball. I think the authors should rephrase that. Btw @authors can you actually see this log difference in experiments on convex objectives? Also, it would be great to cite and compare your result/proof technique with Defossez et al. 2019 (On the convergence of Adam and Adagrad).

5) Many results are presented without proof, such as the ones for the stochastic case.

I think the authors should take some time to work on their interesting direction and resubmit to ICML. I think the results are interesting, just they need to be discussed/complemented/verified a bit better.

------------------------------------------

Score updated after rebuttal, please see comment below

---

> ### Author Response · Authors · 2020-11-17
> **Response to R3**
>
> Thank you for your efforts reviewing our paper and providing helpful comments and suggestions.
>
> A1: To make the assumptions relatively simple and our proof easy to understand, only the concise proof is given in the paper, and other details have already been provided in the original supplementary material.
>
> Thanks for your advice. The momentum parameters can be extended to take the form t/(t+r) for r>2. However, the parameter $\alpha_t$ should be changed accordingly and the derived convergence bound will depend on the parameter r.
>
> A2: (Ghadimi 2015) only consider smooth convex problems and their O(1/t) convergence rate is obviously not optimal (the optimal rate is O(1/t^2) ). As far as we know, we are the first to provide an individual convergence rate for HB which is optimal for nonsmooth convex problems.
>
> From the formulation (7) in the paper, our actual stepsize is still $1/\sqrt t$. Similar to PSG, the projection operation does not affect the stepsize at all.
>
> In nonsmooth convex cases, even for the basic GD, its optimal individual convergence was posed as an open problem on COLT2012 and this open problem remains unsolved until 2019. In our manuscript, we in fact solve the optimal individual convergence of HB. Although there have been some papers discussing the convergence of HB in nonsmooth convex cases, to the best of our knowledge, only averaging convergence is concerned. We believe that it is novel and theoretically shows the power of HB.
>
> We would like to cite (Orvieto et al. 2019) in the revised version. Indeed, there are some other papers considering the increasing momentum strategy. However, as far as know, the relationship between the increasing momentum strategy and optimal individual convergence is not concerned so far.
>
> A3: Yes. We indeed give convergence results in convex cases but conduct the experiments for nonconvex problems. We don’t agree that this does not make sense. In fact, almost all the state-of-art optimizers including the well-known SGD and Adam also suffer from this problem. However, it does not prevent the convex optimization scheme from motivating great empirical successes in deep learning tasks. On the other hand, although the nonconvex problems are recently considered by many papers, their convergence analysis still needs many additional assumptions (such as smoothness) that actually does not hold in deep learning (such as using the activation function ReLU).
>
> (a) We have already considered the hinge loss optimization problem with $l_1$-ball constraints in the original experimental section, which was included the supplementary material.
>
> (b) We follow (Mukkamala & Hein, 2017) and (Wang et al., 2020) to conduct the deep learning experiments. Indeed, decreasing stepsize are also not used there.
>
> (c) In the nonconvex case, there is generally no way to get the expected optimality. In fact, we only prove that a momentum of k/(k+2) is optimal in terms of the individual convergence for nonsmooth convex problems. Similar to SGD, we only provide some motivating hints how the momentum parameters can be scheduled in deep learning. We have compared with a fixed-momentum method, which is denoted as SGD-momentum in our experiment.
>
> A4: Different people have different viewpoints on the understanding of acceleration. From the perspective of theoretical analysis, O(1/t) can naturally be regarded as an acceleration over O(log(t)/t). When comparing with different algorithms with the same order of convergence rate, even when a factor is decreased, it is sometimes called an acceleration in the optimization community.
>
> A concrete example was constructed in (Harvey et al., 2019), where the error of the final iterate of deterministic GD is exactly $\Omega (log(T)/T)$. Thus, we can clearly see this log difference in experiments on this specific example. We have included this example in the experimental section.
>
> We would like to cite (Defossez et al. 2019) in the revised version. It should be indicated they limit the considered nonconvex problem to be smooth while we focus on the nonsmooth convex problem. BTW: for the nonconvex problem, even if the objective function is smooth, we still don’t know how to describe the individual convergence by using the norm of gradient oracle until now.
>
> A5: To make the assumptions relatively simple and our proof easy to understand, we only focus on the deterministic case. At the end of the original introduction, it has already been indicated that the general stochastic and adaptive settings are provided in the supplementary material.
>
> Finally, thanks for your positive evaluation about our work. The only thing we can do now is to try our best to make our manuscript satisfy the level of a ICLR paper, and we are looking forward to your further comments.

---

> > ### Comment · AnonReviewer3 · 2020-11-24
> > **Nice effort, and nice reply**
> >
> > Dear Authors, thanks a lot for your reply.
> >
> > I will raise the score to a weak accept because I saw some effort in improving the paper quality (such as adding the Harvey experiment). The paper changed a bit from the original one, and I think it's way more mature now.
> >
> > Just one thing: I still think going from log(t)/t to 1/t (asymptotically) does not "fill the theory-practice gap" (said in the abstract). This is misleading for a potential reader, please consider removing it: the paper is interesting but I am again not convinced the analysis and the results are strong enough for such a claim. This also because momentum-based methods are also very competitive in the smooth case compared to SGD.
> >
> > All in all, I think the results can be enough for a decent publication in optimization, but the abstract should be modified a bit to match the impact of the results.

---

### Official Review · AnonReviewer1 · 2020-10-28
**The paper needs major rewriting.**

**Rating:** 5
**Confidence:** 4

**Review:**

This paper's major contribution is analyzing the HB method for non-smooth objectives functions and showing the last iterate convergence for it. Comparing to existing results, they show a tighter upper bound by dropping a log(t) term from the
suboptimality's upper bound. It also analysis an adaptive variant of the HB and show similar results. All the proofs are clear and straightforward.

Comments:

1- This paper needs major rewriting and restructuring and in its current format is not ready to be published.
The abstract talks about a specific parameter and its value, which the reader has no idea about it.

 a)- When an acronym is used, its full name should appear once before.

 b)- Typos exist in their proof. For example, in the proof of Lemma 1, and in the last equation, there is no u, but the line above reads for all u in Q. Also, in the paragraph for the paper's 3rd contribution, it says decaying \beta_1,t goes to 1. It is not a decaying but increasing parameter.

c)- There are two appendices, one at the end of the main part, and the other part is in a separate file.

d)- Two related works which haven’t been considered in related work
      Sebbouh O., et.al. On the convergence of the Stochastic Heavy Ball Method
      Sun, T. et. la., Non-Ergodic Convergence Analysis of Heavy-Ball Algorithms

2- In terms of contributions:

a)- I don’t think dropping a log(t) terms in an analysis of a method means acceleration. Usually, Log(t) term appears due to some technical difficulties in the analysis and also in practice when training a model is not a large value.

b)- Since you compare the HB method with SGD, I suggest putting the proof for stochastic HB instead of the deterministic case.

c)- In the experimental section, it would be nice to see the results on bigger datasets like Imagenet and also different tasks such as NLP models. Moreover, for SGD you set the step size \alpha_t to be fixed, which theoretically SGD won’t converge in this situation. To be fair, similar to your sep-size, which is decreasing \alpha_t for SGD should be decreasing with an appropriate rate. Finally, since you run the experiments for 5 runs, it would be useful to add the error bar to your graphs.

---

> ### Author Response · Authors · 2020-11-17
> **Response to R1**
>
> Thank you for your efforts reviewing our paper and providing helpful comments and suggestions. We are looking forward to hearing your response.
>
> A1:
>
> Thanks for your suggestions. We will do our best to revise the manuscript to make it ready to be published. Specifically, we have used momentum parameter to take the place of its specific value.
>
> a)  We have given the full name of SGD.
>
> b) Sorry for the typos, and we have corrected them in the revised version.
>
> c) The content of the first appendix has been moved into Section 3, i.e., the simple but key proof in this paper is included in Section 3, and other details are given in the supplementary material.
>
> d) They restrict the convex problems to be smooth while we focus on the nonsmooth convex problems. What is more, we derive the optimal convergence. Of course, we would like to cite these works in the revised version.
>
>
> A2:
>
> a) A concrete example was constructed in (Harvey et al., 2019), in which the error of the final iterate of deterministic GD is exactly $\Omega (log(T)/T)$. Thus, the effect of acceleration can be clearly illustrated on this specific example. We have included this toy example in the revised experimental section.
>
> We don’t agree that log(t) term appears due to some technical difficulties in the analysis. In fact, it has been proved that the log(t) term is indeed necessary in the convergence rate of GD (Harvey et al., 2019). Thus, how to drop the log(t) term remains a challenging problem in machine learning community.
>
> b) Thanks for suggestions. To make the assumptions relatively simple and our proof easy to understand, we only focus on the deterministic cases. The general stochastic and adaptive settings have been provided in the original supplementary material.
>
> c) Our deep learning experiments are conducted by following a recent ICLR paper (Wang et al., 2020), where only CIFAR10, CIFAR100, MNIST are considered to demonstrate the effectiveness of their methods. Of course, we also would like to see the results on bigger dataset. Unfortunately, our poor hardware conditions prevent us to finish this work in a short time.
>
> We respectfully disagree with the reviewer about the convergence of SGD. In fact, for SGD, even if we suitably choose the time-varying step size $\alpha_t$, the convergence of SGD is still not guaranteed due to the nonconvexity in this situation.
>
> Decreasing $\alpha_t$ for SGD with an appropriate rate is surely a good idea. Unfortunately, it is almost impossible to get this appropriate rate in deep learning tasks. As far as we know, constant $\alpha$ for SGD has been used in almost the deep learning experiments such as (Wang et al., 2020). We have also added the error bar to our graphs.

---

### Author Response · Authors · 2020-11-17
**Paper Revision**

Dear reviewers,

We would like to thank you for all your comments and suggestions. We have uploaded a revised version. The main changes are summarized as follows,

1. A toy example was added in the supplementary material to illustrate the acceleration of HB and AdaHB.

2. The original first appendix has been deleted and its content was moved into Section 3.

We will update this paper again in a few days with further revisions (e.g. add more details about the individual convergence).

---

### Decision · Program_Chairs · 2021-01-07
**Final Decision**

**Decision:**

Accept (Poster)

**Comment:**

This paper studies the *last iterate* convergence of the projected Heavy-ball method (and an adaptive variant) for convex problems, and propose a specific coefficient schedule. All reviewers thought that looking at the last-iterate convergence of the HB method was interesting and that the proofs, while simple, were interestingly novel. Several concerns were raised by the quality of the writing. Several were addressed in a revision and the rebuttal. While R1 did not update their score, the AC thinks that the rebuttal has addressed appropriately their initial concerns. The AC recommends the paper for acceptance, *but* it is important that the authors make an appropriate careful pass over their paper for the camera ready version.

### comments about the write-up

- The paper still contains many typos (e.g. missing $1/t$ term in the average after equation (2); many misspelled words, etc.), please carefully proofread your paper again.
- The AC agrees with R1 that the quality of presentation still needs improvement. $\beta_{1t}$ is still used in the introduction without being defined -- please define it properly first e.g.
- The word "optimal" and "optimality" is usually misused in the manuscript. To refer to the convergence rate of an optimization algorithm, the standard terminology is to talk about the "suboptimality" or the "error" (e.g. see the terminology used by the cited [Harvey et al. 2019, Jain et al. 2019] papers). For example, one would say that the error or suboptimality of SGD has a $O(1/\sqrt{t})$ convergence rate. Saying "optimality of" or "optimal individual convergence rate" is quite confusing, and should be corrected. The adjective "optimal" (when talking about a convergence rate) should be restricted to when a matching lower bound exists.
- Finally, the text introducing the experimental section should be fixed to clarify the actual results and motivation. Specifically, the "validate the correctness of our convergence analysis" only applies in the convex setting. I recommend that a high level description of the convex experiment and the main message of the results is moved from the appendix to the main paper there (there is space). And then, the deep learning experiments can be introduced as just investigating the practical performance of the suggested coefficient schedule for HB.